# Is the Association between Suicide and Unemployment Common or Different among the Post-Soviet Countries?

**DOI:** 10.3390/ijerph19127226

**Published:** 2022-06-13

**Authors:** Nursultan Seksenbayev, Ken Inoue, Elaman Toleuov, Kamila Akkuzinova, Zhanna Karimova, Timur Moldagaliyev, Nargul Ospanova, Nailya Chaizhunusova, Altay Dyussupov

**Affiliations:** 1Department of Psychiatry, Semey Medical University, Semey 071400, Kazakhstan; nurs_7sk@inbox.ru (N.S.); elamantol96@gmail.com (E.T.); akkuzinova@gmail.com (K.A.); jeanna_1997@mail.ru (Z.K.); timur_party@inbox.ru (T.M.); nargul_ospanova@mail.ru (N.O.); 2Research and Education Faculty, Medical Sciences Cluster, Health Service Center, Kochi University, Kochi 780-8520, Japan; 3Department of Public Health, Semey Medical University, Semey 071400, Kazakhstan; n.nailya@mail.ru; 4Chairman of the Board-Rector, Semey Medical University, Semey 071400, Kazakhstan; altay.dyusupov@nao-mus.kz

**Keywords:** USSR, post-Soviet country, suicide, unemployment, male, female

## Abstract

The Union of Soviet Socialist Republics (USSR) collapsed in 1991 and separated into the 15 post-Soviet countries: Armenia, Azerbaijan, Belarus, Estonia, Georgia, Kazakhstan, Kyrgyzstan, Latvia, Lithuania, Moldova, Russia, Tajikistan, Turkmenistan, Ukraine, and Uzbekistan. The post-Soviet countries have faced many economic problems, including unemployment. The association between suicide and unemployment in post-Soviet countries has not been well studied. Here, we researched the annual suicide rate and the unemployment rate during the 28-year period from 1992 to 2019 in the 15 post-Soviet countries. We calculated the correlation coefficients between the suicide rate and the unemployment rate in each of the countries during this period, and we determined the association between the suicide rate and unemployment rate. Our major findings were that (1) the suicide rates among both males and females were significantly associated with the unemployment rate in nearly half of the 15 countries, and (2) for nearly 70% of the males in the entire set of 15 countries, there was an association between the suicide rate and the unemployment rate. Suicide-prevention researchers and organizations should be aware of our findings, and specific suicide-prevention measures based on these results are desirable.

## 1. Introduction

The main reasons that individuals attempt or commit suicide are personal problems, social problems, or combinations of both. According to the World Health Organization, 703,000 people worldwide died as a result of suicide in 2019, and the largest portion of these suicides occurred in low-and middle-income countries, including the independent countries formed as a result of the 1991 collapse of the Union of Soviet Socialist Republics (USSR, or the Soviet Union) [1,2]. The 15 post-Soviet countries are the Republic of Armenia (Armenia), the Republic of Azerbaijan (Azerbaijan), the Republic of Belarus (Belarus), the Republic of Estonia (Estonia), Georgia, the Republic of Kazakhstan (Kazakhstan), the Kyrgyz Republic (Kyrgyzstan), the Republic of Latvia (Latvia), the Republic of Lithuania (Lithuania), the Republic of Moldova (Moldova), the Russian Federation (Russia), the Republic of Tajikistan (Tajikistan), Turkmenistan, Ukraine, and the Republic of Uzbekistan (Uzbekistan) [3]. We will first provide our present major findings regarding the suicide statistics in these individual countries.

Since the 1980s, Armenia has had one of the lowest suicide rates among the post-Soviet countries [4,5,6]. The unemployment rate in the Caucasus countries (i.e., Georgia, Azerbaijan, Armenia, and Russia) has been very high in the last few decades. Georgia’s unemployment rate was highest among the post-Soviet countries from the 1990s to the 2000s, but Armenia’s unemployment rate then became the highest from the mid-2000s to 2016 [7].

A study conducted in Azerbaijan compared the prevalence of suicide among females and males in 2010–2014 using the results of forensic psychiatric examinations [8]. The results revealed that, as in the rest of the world, completed suicides in Azerbaijan were more often committed by males. However, the ratio of suicides by males/females decreased during that study period. We speculated that a potential reason for the finding of a decrease ratio is that the study used two age groups (21–30 years and ≥60 years) for females but a single age group (41–50 years) for males [8].

Another investigation divided the countries of the former USSR into clusters according to the historically established suicide rate in the countries, and the subsequent analysis demonstrated that the link with alcohol consumption was most clearly visible in countries with high suicide rates [9]. This relationship between alcohol consumption and suicide is another reflection of the impact of psychosocial stress.

At the time of the collapse of the USSR, Belarus chose the path of less radical socio-economic reforms [10]. In Georgia, according to the country’s National Statistics Service, the unemployment rate gradually decreased from 2009 to 2017 [11]. Most of the unemployed individuals in Georgia are in the age groups 20–24 years and 25–29 years [12]. Georgia’s unemployment rate trend continues to be higher than the world’s average, and thus a large portion of the Georgian population may face a loss of jobs and income; the stress due to this possibility is thus likely to be high.

In our previous study, we calculated the suicide rate in Kazakhstan from 2000 to 2019 and examined several items reflecting labor, financial, and economic factors by conducting a multiple regression analysis [13]. The results revealed that only unemployment was significantly associated as a risk factor for suicide during that 20-year period. Our findings also indicated that further long-term studies in the post-Soviet countries are needed and that it is important to analyze countries that are similar to Kazakhstan.

Kyrgyzstan in the post-Soviet period has been characterized by an uneven increase in the country’s unemployment rate, especially among females (due to the society’s structure) [14]. According to a report from the United Nations Population Fund (UNFPA), approximately one-quarter of the suicide attempts in Kyrgyzstan in 2012 were committed by individuals aged 18–22 years [15], and the rates of completed suicide in those aged 15–29 years were high (16.3 for males and up to 6.6 for females per 100,000 population) [16].

In the 1990s, the countries of Eastern Europe experienced significant changes in the socio-economic sphere caused by the collapse of the USSR [17]. Some of the countries in Eastern Europe that were most significantly affected in socio-economic terms by the collapse of the Soviet Union were the Baltic states, i.e., Latvia, Lithuania, and Estonia. The accompanying dynamics of the suicide rate in these three countries are very similar, showing high suicide rates in the 1990s [17,18,19]. The Baltic states have continued to have high suicide rates [1,4]. In fact, suicide has long been among the top five causes of death in Latvia. In the 1990s, this was due in part to the economic difficulties experienced by Latvia during the period of its new independence, and these difficulties were important factors contributing to the increase in the country’s number of suicides. The steady growth of Latvia’s gross domestic product (GDP) since 1980 and the subsequent halving of the GDP from 1990 to 1993 occurred as a result of major changes in the Latvian economic system due to the country’s secession from the USSR [18]. That study also identified an important finding: a drop in the indicator GDP was accompanied by an increase in the number of suicides in Latvia [18].

In recent decades, Lithuania has experienced significant social, political, and economic changes associated with the change in the status of the country from one of the most developed republics of the Soviet Union to an independent state that continues to attempt to develop its market economy. The population has thus faced an unfamiliar social environment and has experienced stress resulting from the changes [20]. Periods of serious socio-economic activity instability (from 1990 to 1994, when Lithuania was just beginning to become a separate and independent state, and the global financial crisis in 2008) were accompanied by sharp increases in the country’s suicide rate [21,22].

In Moldova, it was reported that the suicide rate was at almost the worldwide average level throughout the early 1990s to the early 2010s [23]. Kõlves et al. reported that the suicide rates among males in Moldova remained nearly stable from 1990 to 2008 [24]. Russia has had a very high suicide rate compared to the rest of the world, and the rank is still high in the post-Soviet era. One of the proposed reasons for this high rank is the high level of alcoholism in Russia [25].

The socio-economic situation in Tajikistan in the ~30 years since the collapse of the Soviet Union has been characterized by rapid population growth, a labor surplus, and massive labor migration. The greatest portion of the working-age population consists of young people, and their share among the unemployed is growing [26]. A study of suicide attempts and completed suicides from 2005 to 2010 in Tajikistan showed that the greatest number of suicides was committed by individuals aged 15–29 years; suicides among females engaged in domestic work have been predominant, and employed and unemployed persons make up roughly equal percentages [27].

Turkmenistan has had one of the lowest rates of suicide, along with Uzbekistan and Tajikistan, since 1991 [28,29]. The reasons for the low suicide rate in Turkmenistan remain to be clarified.

The causes of the dependence of mortality in Ukraine on a serious lung disease, i.e., tuberculosis, and the causes of suicide in Ukraine have been examined [30]; it was observed that a common reason for suicide was psychosocial distress caused by the country’s socio-economic crisis and a sharp drop in the population’s standard of living. The greatest exposure was among the least protected segments of society, and Ukrainians with reduced social adaptation were the most susceptible to intense psycho-emotional influences [30].

The importance of preventing widespread unemployment and that of suicide-prevention measures among younger people in Uzbekistan have been described [31]. The author of that study noted that in order to resolve the imbalance between supply and demand in the labor market, it is necessary to know what specialists are needed and in what quantity they are needed for the development of the economy in the present and in the future; what labor skills and abilities specialists should have; and the necessity of training for young people in specialties commissioned by enterprises. The study’s author further noted that strengthening the success of labor adaptation in a new workplace depends on the personal qualities, business qualities and mobility of individuals, and on the level of their professional skills acquired in their university studies [31]. Investigations of the suicide rate in Estonia suggested that the rate may depend on socio-political, economic, and psychological factors [32,33]. Estonia’s suicide rate was reported to be lower than those in the other two Baltic States but higher than those of many countries in Europe and the former USSR [4,5].

When the Soviet Union ceased to exist, the above-described countries faced many economic problems, and one of the major issues has been how to prevent unemployment [7]. Unemployment can lead to economic and life problems that contribute to motives for suicide, and unemployment has been associated with suicide in both Japan and South Korea [34,35]. There are many triggers for suicide, and unemployment is a potential trigger that merits greater attention [36,37]. As a global problem, the prevention of suicide is a pressing public health issue worldwide, including the post-Soviet countries. It is important to determine the precise association between suicide and unemployment in post-Soviet countries. Our present analyses were designed to answer the following question: Is the association between suicide and unemployment common or different among the post-Soviet countries?

## 2. Materials and Methods

### 2.1. Data

We obtained the annual suicide rate during the 28-year period from 1992 to 2019 among the total population, males, and females in each of the 15 post-Soviet countries from the Global Health Data Exchange [38]. These countries are Armenia, Azerbaijan, Belarus, Estonia, Georgia, Kazakhstan, Kyrgyzstan, Latvia, Lithuania, Moldova, Russia, Tajikistan, Turkmenistan, Ukraine, and Uzbekistan. We obtained the annual unemployment rate each of the countries during the same period from the International Labour Organization (ILO) at the website of the GLOBAL NOTE company, which specializes in official international statistical data in various fields [39].

### 2.2. Statistical Analysis

During the study period, we calculated the correlation coefficients between the suicide rate and the unemployment rate in the 15 post-Soviet countries. The association between the suicide rate and the unemployment rate was examined by a simple regression analysis in the Excel program.

### 2.3. Interpretation

The results of the above-described statistical analyses indicated that unemployment was a common risk factor for suicide in each of the 15 post-Soviet countries. The details of this result are described next.

## 3. Results

### 3.1. The Annual Suicide Rate and Unemployment Rate in the 15 Post-Soviet Countries during the 28-Year Period 1992–2019

As illustrated in Figure 1, during the 28-year study period in the 15 above-cited countries after the dissolution of the Soviet Union, the annual suicide rates (per population of 100,000) from the minimum to the maximum in the total population were as follows. Armenia 2.10–7.40, Azerbaijan 2.44–4.90, Belarus 21.20–44.30, Estonia 14.90–45.50, Georgia 5.37–10.70, Kazakhstan 17.60–39.70, Kyrgyzstan 7.40–15.08, Latvia 18.10–47.73, Lithuania 26.10–50.17, Moldova 14.40–22.16, Russia 25.10–56.37, Tajikistan 3.40–4.45, Turkmenistan 5.70–15.10, Ukraine 20.60–39.10, and Uzbekistan 7.05–10.10.

Among only the males during the same study period, the annual suicide rates (per population of 100,000) from the minimum to the maximum were as follows (Figure 2): Armenia 3.40–10.90, Azerbaijan 3.89–7.80, Belarus 36.70–80.10, Estonia 24.30–77.90, Georgia 8.56–18.60, Kazakhstan 29.00–69.20, Kyrgyzstan 11.70–25.00, Latvia 34.30–81.71, Lithuania 45.40–86.80, Moldova 26.00–36.70, Russia 43.60–98.70, Tajikistan 4.70–6.35, Turkmenistan 8.80–25.00, Ukraine 37.20–70.90, and Uzbekistan 10.53–16.00.

As depicted in Figure 3, among the females during the same study period, the annual suicide rates (min.–max.) were: Armenia 0.70–5.20, Azerbaijan 1.04–2.10, Belarus 7.70–12.90, Estonia 5.40–17.39, Georgia 2.19–3.60, Kazakhstan 6.80–12.40, Kyrgyzstan 3.20–6.41, Latvia 4.30–18.29, Lithuania 9.60–18.40, Moldova 3.80–8.84, Russia 9.10–19.00, Tajikistan 1.99–3.10, Turkmenistan 2.70–6.00, Ukraine 6.30–13.57, and Uzbekistan 3.62–5.70.

The minimum–maximum annual unemployment rates (%) during the study period were: Armenia 1.80–19.01, Azerbaijan 1.80–11.80, Belarus 3.10–24.40, Estonia 3.68–16.71, Georgia 5.40–20.71, Kazakhstan 1.00–13.46, Kyrgyzstan 1.10–12.55, Latvia 6.05–20.70, Lithuania 1.20–17.81, Moldova 2.98–11.14, Russia 4.59–13.26, Tajikistan 2.10–16.50, Turkmenistan 1.50–12.70, Ukraine 1.90–11.86, and Uzbekistan 2.90–13.30 (Figure 4).

### 3.2. The Correlation between the Suicide Rate and the Unemployment Rate in the Post-Soviet Countries in 1992–2019

The correlation coefficients between the suicide rates among the total population, males, and females and the unemployment rate in the 15 post-Soviet countries during the study period are provided in Table 1. There were seven countries in which the suicide rates in males, females, and the total population each showed a significant positive correlation with the unemployment rate: Armenia, Belarus, Georgia, Kazakhstan, Latvia, Lithuania, and Russia.

In contrast, the suicide rates among males and the total population in Uzbekistan were significantly positively correlated with the unemployment rate, but the suicide rate among females was significantly inversely correlated with the unemployment rate. Two other countries in which the suicide rates among males and among the total population were significantly positively correlated with the unemployment rate were Moldova and Turkmenistan.

In Azerbaijan and Kyrgyzstan, the suicide rate of females was significantly inversely correlated with the unemployment rate. In Tajikistan, the suicide rates in males, females, and total population were each significantly inversely correlated with the unemployment rate.

The two countries in which the suicide rates of the total population, males, and females were not correlated with the unemployment rate were Estonia and Ukraine.

### 3.3. The Association between the Suicide Rate and the Unemployment Rate in the Populations of the 15 Post-Soviet Countries

Table 2 provides the data of the 15 countries’ associations between the suicide rate in the total population, males, and females and the unemployment rate, plus the regression lines. In the seven countries Armenia, Belarus, Georgia, Kazakhstan, Latvia, Lithuania, and Russia, the suicide rates among both males and females and in the total population were significantly associated with the unemployment rate.

In Uzbekistan, the suicide rates among males and in the total population were significantly associated with the unemployment rate, and the suicide rate among females was significantly inversely associated with the unemployment rate.

The two countries in which the suicide rate among males and in the total population were significantly associated with the unemployment rate were Moldova and Turkmenistan.

In Azerbaijan and Kyrgyzstan, a significant inverse association was observed between the suicide rate in females and the unemployment rate, whereas in Tajikistan, the suicide rates among males, females, and the total population were significantly inversely associated with the country’s unemployment rate.

The two countries in which the total suicide rate and the rates in the males and females were not associated with the unemployment rate were Estonia and Ukraine.

## 4. Discussion

Based on our present findings, particularly those described above in Section 3.2 and Section 3.3, we observed three groups that describe the association between suicide and the unemployment in the 15 post-Soviet countries. In group 1, there were associations between the suicide rate and the unemployment rate among both males and females: Armenia, Belarus, Georgia, Kazakhstan, Latvia, Lithuania, and Russia. In group 2, there was an association between suicide and unemployment only in males: Moldova, Turkmenistan, and Uzbekistan. In group 3, no association was revealed between suicide and unemployment: Azerbaijan, Estonia, Kyrgyzstan, Tajikistan, and Ukraine.

There are some countries in groups 2 and 3 in which the suicide rate was associated with the unemployment rate as a significant inverse correlation. The analysis results indicated that (*i*) the suicide rate increased with decreasing unemployment rate, or (*ii*) the suicide rate decreased with increasing unemployment rate. This tendency was not realistic and was incomprehensible. We speculated that our finding that the suicide rate was associated with the unemployment rate as a significant inverse correlation actually meant that there was no correlation or association between the countries’ suicide rate and unemployment rate. In Tajikistan only, the suicide rate and the unemployment rate in the total population showed a strong negative correlation. There have been few detailed investigations of suicide in Tajikistan to date, and we were unable to find any previous studies suggesting a reason for this relationship. It is necessary to further examine this relationship by investigating more aspects of economic/political events, climate, culture, and other factors.

The group 1 criterion, i.e., an association between suicide and unemployment among both males and females, was observed in nearly half (seven) of the 15 post-Soviet countries. In reports from Japan, Spain, and Italy, the unemployment rate was more closely associated with male suicides than female suicides [40,41,42]. Therefore, in post-Soviet countries an association between suicide and unemployment among both males and females may be one of the features of suicide due to the unemployment. It is possible that the employment rate of females in the USSR was high during the USSR’s existence, and that employment was thus an important item in the economic and daily lives of not only males, but also females.

The association between suicide and unemployment among males (in the present countries in group 2) is common worldwide. Nordt et al. reported that the 2008 economic crisis had the most significant impact on the growth of unemployment, and males carried out more suicide efforts than females [43]. In the present study, group 2 was comprised of three (20%) of the 15 post-Soviet countries.

The absence of any association between suicide and unemployment (group 3, five countries) was observed in approximately 30% of the 15 post-Soviet countries. We suspect that factors other than unemployment were closely associated with suicide in these countries. However, this trend may change in the future, and continued research is thus necessary.

Unemployment can lead to mental fatigue, a depressive state, or psychiatric disorders, all of which can lead to suicide [44,45,46]. Our present analyses uncovered two major points regarding the features of suicide linked to unemployment in many of the post-Soviet countries: (1) the suicide rates of both males and females were significantly associated with the unemployment rate in nearly half of the 15 countries, and (2) for nearly 70% of the males in the entire set of 15 countries, there was a significant association between the suicide rate and the unemployment rate. Public and private organizations and researchers involved in suicide prevention should be aware of these two points in both the entire set of 15 post-Soviet countries and in their individual countries.

There are several study limitations to note. The definition of ‘unemployed’ according to the ILO is ‘those who are older than a certain age, have no work in a specific period, can work during that period, and have been looking for a job in the most recent period’ [47]. Individual countries have defined ‘unemployed’ in accordance with the ILO standard, but the content of the definitions differs slightly. Our present findings were based on the association between suicide rates and unemployment rates and do not define causality. The subject countries were also limited to the former USSR. Our results thus provide a starting point for discussions about economic-based risk factors for suicide based on statistical analyses using only numerical data without individual information. We did not examine the backgrounds of individual suicides.

In addition, the correlation coefficients and single R^2^ values were not great in this study. Although in many circumstances there may be statistically significant positive associations between suicide and unemployment, the associations may be in general not very strong. Moreover, we were able to obtain and analyze annual data in this study, and we want to conduct further analyses using data with higher temporal granularity in the future.

Although the USSR collapsed 30 years ago, we have found no prior report about trends of the association between suicide and unemployment in any of the post-Soviet countries or whether the association is common, similar, or different among these 15 countries. Future investigations should apply additional statistical approaches including ecological analyses, and it is important to compare the existing findings with their results. For example, Émile Durkheim’s theory of suicide focuses on a group-minded collective consciousness, and it examines four types of suicide (egoistic suicide, altruistic suicide, anomic suicide, and fatalistic suicide) [48]. We speculate that our present findings may be linked to egoistic suicide in particular. It is also necessary to discuss new viewpoints based on the Durkheim theory in more precise studies of suicide in individual countries. Our present findings provide new data and viewpoints toward this goal.

Two major issues (the COVID-19 pandemic and the Russia/Ukraine conflict) are ongoing as of this writing, and these issues may well cause isolation, depression, and trauma among the people in the 15 post-Soviet countries. A continuing dialog concerning suicide in the 15 post-Soviet countries is necessary.

## 5. Conclusions

Our analyses revealed three groups among the 15 post-Soviet countries based on the association between the rate of suicide and the unemployment rate. The major findings are (*i*) the suicide rates of both males and females were significantly associated with the unemployment rate in nearly half of the 15 countries; and (*ii*) for nearly 70% of the males in the entire set of 15 countries, there was a significant association between the suicide rate and the unemployment rate. Point (*i*) in particular was a feature among the 15 post-Soviet countries. Suicide researchers and the staff of suicide prevention organizations should be aware of the trends in both their own countries and in the 15 post-Soviet countries as a whole. Entities in the suicide-prevention field including administration, labor institutions, and medicine should discuss further research and specific prevention measures together as necessary. If these entities and individuals become familiar with the present situation regarding the widespread increase in unemployment, they will be aware that (*i*) the rate of suicide may be increasing in their country, and (*ii*) many individuals in their country may be suffering from increased fatigue and poor physical and mental health. Such realizations could contribute to more effective suicide prevention measures. Suicide is a complex issue, and we believe that maintaining a wide perspective (including the content of this report) may be help identify new and effective suicide prevention measures.

## Figures and Tables

**Figure 1 ijerph-19-07226-f001:**
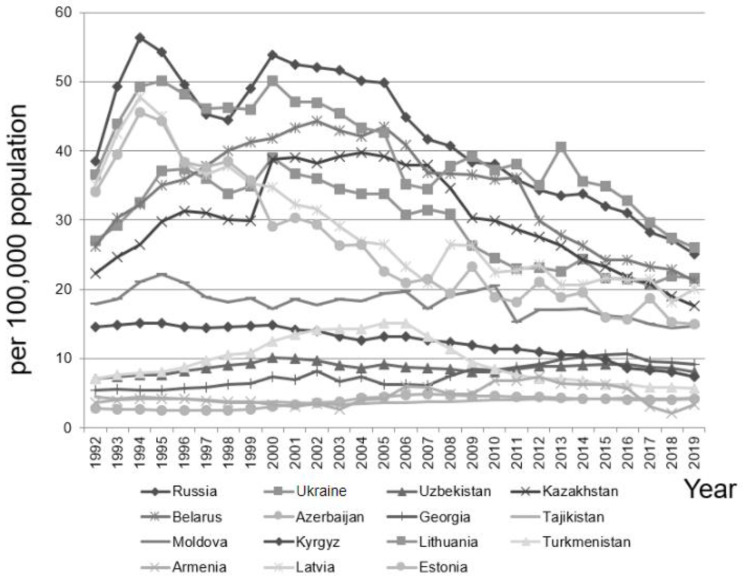
The annual suicide rate among all individuals in the 15 post-Soviet countries during the 28-year period from 1992 to 2019.

**Figure 2 ijerph-19-07226-f002:**
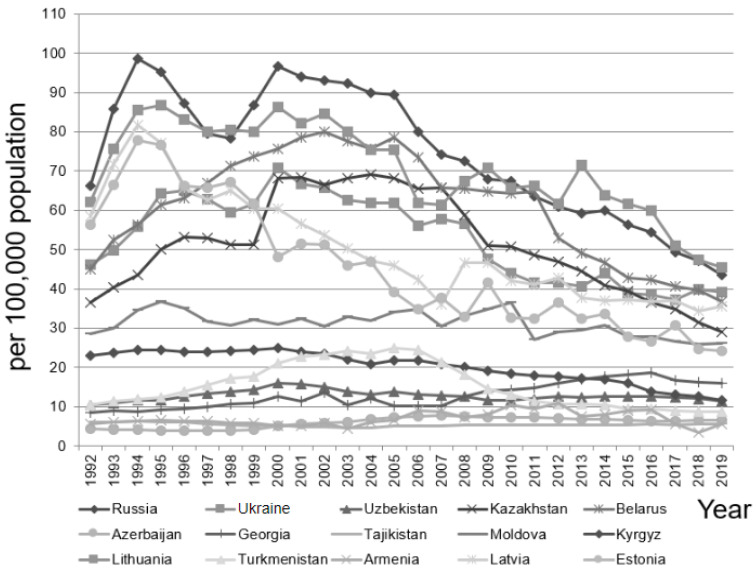
The annual suicide rate among males in the 15 post-Soviet countries in 1992–2019.

**Figure 3 ijerph-19-07226-f003:**
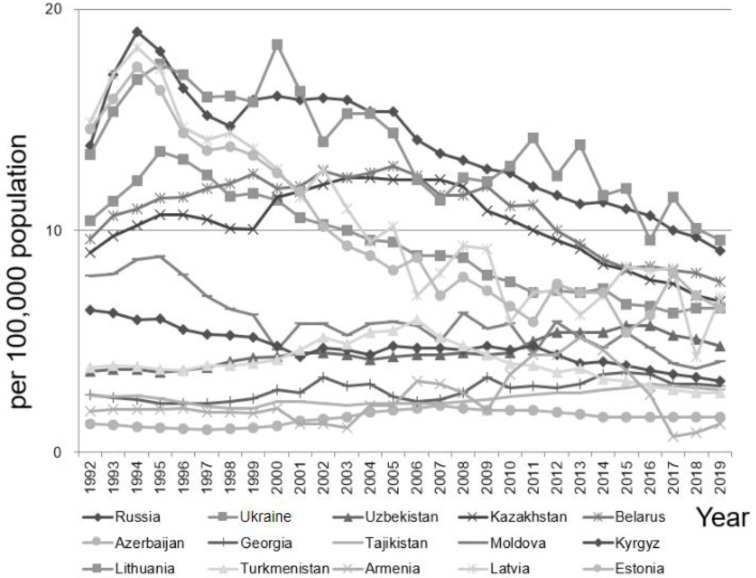
The annual suicide rate among females in the 15 post-Soviet countries in 1992–2019.

**Figure 4 ijerph-19-07226-f004:**
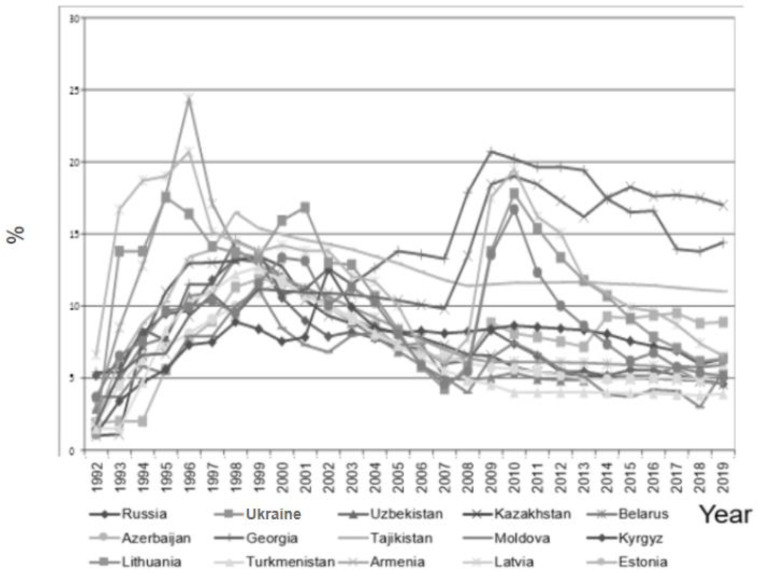
The annual unemployment rate in the 15 post-Soviet countries in 1992–2019.

**Table 1 ijerph-19-07226-t001:** Correlations between the suicide rate and unemployment rate in each of the 15 post-Soviet countries.

	Total Population r-Value, *p*-Value	Males r-Value, *p*-Value	Females r-Value, *p*-Value
Armenia	0.425, 0.024 *	0.436, 0.021 *	0.382, 0.045 *
Azerbaijan	−0.359, 0.061	−0.340, 0.077	−0.382, 0.045 *
Belarus	0.468, 0.012 *	0.452, 0.016 *	0.502, 0.007 **
Estonia	0.170, 0.387	0.198, 0.314	0.059, 0.765
Georgia	0.709, 2.5 × 10^−5^ ***	0.712, 2.2 × 10^−5^ ***	0.536, 0.003 **
Kazakhstan	0.565, 0.002 **	0.572, 0.001 **	0.490, 0.008 **
Kyrgyzstan	−0.075, 0.703	−0.012, 0.952	−0.419, 0.026 *
Latvia	0.598, 7 × 10^−4^ ***	0.630, 3.23 × 10^−4^ ***	0.492, 0.008 **
Lithuania	0.688, 5.2 × 10^−5^ ***	0.689, 5 × 10^−5^ ***	0.677, 7.6 × 10^−5^ ***
Moldova	0.377, 0.048 *	0.453, 0.015 *	0.114, 0.563
Russia	0.647, 1.98 × 10^−4^ ***	0.645, 2.14 × 10^−4^ ***	0.633, 3 × 10^−4^ ***
Tajikistan	−0.701, 3.3 × 10^−4^ ***	−0.659, 1.37 × 10^−4^ ***	−0.466, 0.012 *
Turkmenistan	0.598, 7 × 10^−4^ ***	0.625, 3.8 × 10^−4^ ***	0.352, 0.066
Ukraine	0.096, 0.628	0.158, 0.421	−0.179, 0.361
Uzbekistan	0.492, 0.008 **	0.750, 4 × 10^−6^ ***	−0.394, 0.038 *

** p* < 0.05, ** *p* < 0.01, *** *p* < 0.001.

**Table 2 ijerph-19-07226-t002:** The association between the suicide rate and the unemployment rate, and the regression lines in the simple regression analysis.

	Total R^2^-Value *p*-Value	Males R^2^-Value *p*-Value	Females R^2^-Value *p*-Value	Total y=	Males y=	Females y=
Armenia	0.181 <0.05 *	0.190 <0.05 *	0.146 <0.05 *	0.1281x + 2.940	0.1703x + 4.792	0.0965x + 1.178
Azerbaijan	0.129 >0.05	0.116 >0.05	0.146 <0.05 *	−0.1263x + 4.592	−0.1921x + 7.279	−0.5302x + 1.911
Belarus	0.219 <0.05 *	0.204 <0.05 *	0.252 <0.01 **	0.7187x + 27.691	1.3038x + 48.977	0.1729x + 9.279
Estonia	0.029 >0.05	0.039 >0.05	0.003 >0.05	0.4963x + 22.018	0.9978x + 36.478	0.0668x + 9.386
Georgia	0.502 <0.001 ***	0.507 <0.001 ***	0.287 <0.01 **	0.2928x + 3.543	0.5427x + 5.318	0.0554x + 2.047
Kazakhstan	0.319 <0.01 **	0.327 <0.01 **	0.240 <0.01 **	1.1148x + 21.601	2.0690x + 35.601	0.2409x + 8.420
Kyrgyzstan	0.006 >0.05	1 × 10^−4^ >0.05	0.175 <0.05 *	−0.0860x + 13.019	−0.0232x + 20.399	−0.1686x + 5.981
Latvia	0.358 <0.001 ***	0.397 <0.001 ***	0.242 <0.01 **	1.1521x + 14.278	1.9729x + 25.338	0.4349x + 5.058
Lithuania	0.473 <0.001 ***	0.475 <0.001 ***	0.458 <0.001 ***	1.0742x + 28.054	1.8486x + 49.551	0.3729x + 9.637
Moldova	0.142 <0.05 *	0.205 <0.05 *	0.013 >0.05	0.3608x + 15.790	0.6846x + 26.950	0.0765x + 5.420
Russia	0.419 <0.001 ***	0.415 <0.001 ***	0.401 <0.001 ***	2.4568x + 23.733	4.4152x + 41.477	0.6895x + 8.673
Tajikistan	0.491 <0.001 ***	0.434 <0.001 ***	0.217 <0.05 *	−0.0760x + 4.840	−0.1013x + 6.611	−0.0522x + 3.073
Turkmenistan	0.358 <0.001 ***	0.390 <0.001 ***	0.124 >0.05	0.5723x + 5.982	1.0602x + 8.582	0.0915x + 3.486
Ukraine	0.009 >0.05	0.025 >0.05	0.032 >0.05	0.2202x + 27.747	0.6464x + 47.640	−0.1529x + 10.578
Uzbekistan	0.242 <0.01 **	0.562 <0.001 ***	0.155 <0.05 *	0.1245x + 7.752	0.3452x + 10.337	−0.0856x + 5.147

Response variable (y): The suicide rate. Explanatory variable (x): the unemployment rate (x). * *p* < 0.05, ** *p* < 0.01, *** *p* < 0.001.

## Data Availability

Not applicable.

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
