# Peer review of "Is the Association between Suicide and Unemployment Common or Different among the Post-Soviet Countries?"

_ijerph, 2022, doi:10.3390/ijerph19127226_

Round 1
Reviewer 1 Report
This paper presents an analysis of associations in post-Soviet countries between suicide and unemployment. The analysis is simple but offers a potential launch point for additional follow-on work. Overall, I thought the paper is reasonable, but I have some comments:
- There are several typos in the paper. I suggest the authors run the paper through a spellchecker. For example, here's is a non-exhaustive list:
- last sentence of abstract: "speicific" should be "specific"
- Figure 1's caption: "oveall" should be "all"
- Table 1 and 2: "Ukraina" should be "Ukraine"
- Paragraph 2 of the Introduction states that the authors suspect that the reason that Armenia has some of the lowest suicide rates is due to the fact that family ties are very strong, and people tend to live with their large families.
- Are family ties and living situations different in Armenia than in the other 14 countries?
- Has this effect (the hypothesized positive effect of family and living situations) been shown in other regions of the world to reduce suicide rates?
- The paper analyzes data from 1992-2019, which obviously excludes the COVID-19 pandemic and current Ukraine/Russia crisis. I feel the authors should mention these two current issues (and possibly others) as grounds for more detailed follow-on studies that include data since 2019. I can't help but wonder how suicide has manifested given these incredibly important (and stressful) issues.
- Tajikistan is particularly interesting in the authors' results; it's the only country with a strong negative correlation (r = -0.701, very small p). The authors barely mention this, and I feel that it warrants a more detailed discussion. Why is Tajikistan so different? What about the economic/political climate or culture (or other factors) might contribute to this difference?
- Did the authors consider a lagged analysis? For example, a 1- or 2-year lag might be interesting to look at because unemployment rates might have a delayed impact on suicide.
- Did the authors look at the first derivative of the rates (i.e., the change in the rates between successive years)? I imagine that what might play a more impactful role than unemployment itself is actually the change in unemployment through time. If there's a sharp increase in unemployment from year 1 to year 2, then that might play a more important role in suicide than sustained unemployment rates.
- In the Results, Discussion, and Conclusion sections, the authors continually refer to the fact that they identified significant correlations. While this is true (i.e., due to very small p values), many of the correlations in Table 1 and the R2 values in Table 2 are not very strong, and I feel like this language conveys some assurance in the associations that's not actually present in the output. For example, in Table 1, while Latvia's and Uzbekistan's p values convey strong statistical significance (p < 0.001 and p < 0.01, respectively), the correlation coefficients aren't great (r = 0.598 and r = 0.492, respectively); this problem is exacerbated in Table 2, where there's not a single R2 value greater than 0.56. While there may be positive associations that are statistically significant between suicide and unemployment in many circumstances explored by the authors, the associations are in general not very strong. Did the authors have a cutoff in mind for what constitutes a more trusted correlation coefficient or trusted R2 value? I think it would be beneficial for the authors to explicitly state and justify how r and R2 values are interpreted in the paper.
- The authors analyze annual data. If the authors could analyze monthly or weekly data, do they have any intuition in what might be discovered? This might be nice to mention as future work (i.e., analyze this same effect using data with higher temporal granularity).
Author Response
Reviewer 1:
This paper presents an analysis of associations in post-Soviet countries between suicide and unemployment. The analysis is simple but offers a potential launch point for additional follow-on work. Overall, I thought the paper is reasonable, but I have some comments:
Response: Thank you very much for your comments. The revised text is presented in red (Reviewer 1), purple (Reviewer 2), and green (Reviewer 3) font in yellow highlighting.
There are several typos in the paper. I suggest the authors run the paper through a spellchecker. For example, here's is a non-exhaustive list:
last sentence of abstract: "speicific" should be "specific"
Figure 1's caption: "oveall" should be "all"
Table 1 and 2: "Ukraina" should be "Ukraine"
Other: Figures 1-4: "Ukraina" to "Ukraine"
Response: We have corrected all of the spelling errors.
Paragraph 2 of the Introduction states that the authors suspect that the reason that Armenia has some of the lowest suicide rates is due to the fact that family ties are very strong, and people tend to live with their large families.
Are family ties and living situations different in Armenia than in the other 14 countries?
Has this effect (the hypothesized positive effect of family and living situations) been shown in other regions of the world to reduce suicide rates?
Response: We discussed the text that you’re describing and concluded that some parts were exaggerated; they were corrected as follows.
“Since the 1980s, Armenia has had one of the lowest suicide rates among the post-Soviet countries [4–6].”
The paper analyzes data from 1992-2019, which obviously excludes the COVID-19 pandemic and current Ukraine/Russia crisis. I feel the authors should mention these two current issues (and possibly others) as grounds for more detailed follow-on studies that include data since 2019. I can't help but wonder how suicide has manifested given these incredibly important (and stressful) issues.
Response: We added the contents in the last paragraph of the Discussion as follows.
“Two major issues (the COVID-19 pandemic and the Russia/Ukraine conflict) are ongoing as of this writing, and these issues may well cause isolation, depression, and trauma among the people in the 15 post-Soviet countries. A continuing dialog concerning suicide in the 15 post-Soviet countries is necessary.”
Tajikistan is particularly interesting in the authors' results; it's the only country with a strong negative correlation (r = -0.701, very small p). The authors barely mention this, and I feel that it warrants a more detailed discussion. Why is Tajikistan so different? What about the economic/political climate or culture (or other factors) might contribute to this difference?
Response: We added the contents in the Discussion section’s 2nd paragraph, last sentence as follows.
“In only Tajikistan, the suicide rate and the unemployment rate in the total population showed a strong negative correlation. There have been few detailed investigations of suicide in Tajikistan until now, and we were unable to find any previous studies suggesting a reason for this relationship. It is necessary to further examine this relationship by investigating more aspects of economic/political events, climate, culture, and other factors.”
Did the authors consider a lagged analysis? For example, a 1- or 2-year lag might be interesting to look at because unemployment rates might have a delayed impact on suicide.
Did the authors look at the first derivative of the rates (i.e., the change in the rates between successive years)? I imagine that what might play a more impactful role than unemployment itself is actually the change in unemployment through time. If there's a sharp increase in unemployment from year 1 to year 2, then that might play a more important role in suicide than sustained unemployment rates.
Response: We had discussed the study of this issue from various angles: “1. In our discussion, we reviewed methods about studies of this kind in the world.” and “2. In the present situation of suicide related reports in the 15 post-Soviet countries, we confirmed that this report should be published as our first report.” We thus performed the statistical analysis of our findings.
We think that your advice is important, and we have therefore added the following to the 9th paragraph of the Discussion.
“Future investigations should apply additional statistical approaches including ecological analyses, and it is important to compare the existing findings with their results.”
In the Results, Discussion, and Conclusion sections, the authors continually refer to the fact that they identified significant correlations. While this is true (i.e., due to very small p values), many of the correlations in Table 1 and the R2 values in Table 2 are not very strong, and I feel like this language conveys some assurance in the associations that's not actually present in the output. For example, in Table 1, while Latvia's and Uzbekistan's p values convey strong statistical significance (p < 0.001 and p < 0.01, respectively), the correlation coefficients aren't great (r = 0.598 and r = 0.492, respectively); this problem is exacerbated in Table 2, where there's not a single R2 value greater than 0.56. While there may be positive associations that are statistically significant between suicide and unemployment in many circumstances explored by the authors, the associations are in general not very strong. Did the authors have a cutoff in mind for what constitutes a more trusted correlation coefficient or trusted R2 value? I think it would be beneficial for the authors to explicitly state and justify how r and R2 values are interpreted in the paper.
The authors analyze annual data. If the authors could analyze monthly or weekly data, do they have any intuition in what might be discovered? This might be nice to mention as future work (i.e., analyze this same effect using data with higher temporal granularity).
Response: We added the following text to the 8th paragraph of the Discussion.
“In addition, the correlation coefficients and single R2 values were not great in this study. Although in many circumstances there may be statistically significant positive associations between suicide and unemployment, the associations may be in general not very strong. Moreover, we were able to obtain and analyze annual data in this study, and we want to conduct further analyses using data with higher temporal granularity in the future.”
Other
Response: The References list has been updated with new references and with ‘doi’ information.

Reviewer 2 Report
This is a systematic review about suicidal rate among post-soviet countries.
[Section 2.2: Statistical analyses]: please explain better statistical analysis, and if your data presents a normal distribution or not and why you use Pearson's correlation method.
Please, read the latex-form (Word) provided by MDPI, on the references section is demanded to insert DOI
Author Response
Reviewer 2:
This is a systematic review about suicidal rate among post-soviet countries.
Response: Thank you very much for your comments, which have helped us improve our manuscript. The revised text is presented in red (Reviewer 1), purple (Reviewer 2), and green (Reviewer 3) font in yellow highlighting.
[Section 2.2: Statistical analyses]: please explain better statistical analysis, and if your data presents a normal distribution or not and why you use Pearson's correlation method.
Response: We understood that these data presented a normal distribution, and we also performed the analysis using Excel. Based on your comments, we reconsidered the descriptions and now show the corrections in the section ‘2.2 Statistical analysis’ as follows. We also corrected the numerical value in Table 2 (red font); the meaning of the results is not changed.
“During the study period, we calculated the correlation coefficients between the suicide rate and the unemployment rate in the 15 post-Soviet countries. The association between the suicide rate and the unemployment rate was examined by a simple regression analysis in the Excel program."
Please, read the latex-form (Word) provided by MDPI, on the references section is demanded to insert DOI
Response: We have added the ‘doi’ information that could be found.
Other
Response: The References list has been updated with new references and with ‘doi’ information.

Reviewer 3 Report
This is an interesting study, shows social problems. But I don’t understand the main goal of study.
You decided to present correlational research. In your opinion joblessness is an interesting spectrum to observe fluctuation of suicidal rate.
Ok, but why unemployment? It is the same like different ones; heartbreak, impoverished relations, to be a victim of crime, homelessness, divorce, become widowed, debts and so on.
Following Your text we are witness of „figures playing”, unemployment – suicide - a few counties. The rates change in period of time, thats all. It is not a new poin of viwe.
That concpetion is quite obvious. In all countries, during structural, social and political changes, we are able to observe fluctuation of suicide.
Question 1
Have you got any theoretical model of research? Accordance to Durkheim- there are facts (statstics), so what about argue.
Do you have any theoretical reason to take off „unemployment” as a specific (remarkable) factor for suicide? Generally speaking- are You able to explain your model of research in theories of suicide?
Question 2.
Are You able to explain your poin of view? How do You consider suicide as personal and social problem? Much as I admire your invention. But as far me that picture needs more knowlege about process of suicide (not as a suicide rate).
What is the first:
Do You think unemployment there is motive of suicide, or only a trigger of it? What is real couse?
A lot of theories have pointed out unemployment only as a trigger. It corresponds with personal featuers, patrimonial tendency.
Accordance my research (goodby letters and personal story) there is a tendecy:
Childeren- school and relatiions’ problems there are triggers of self-destructive behaviour
Youth- love and heartberak
Adults- joblessnes, debts, divorce, bankruptcy
Old people- become widowed, lonelines feeling, physical suferring
Look, it changes in part of life, but the cause it means twine (web) of problems, and decision evolve in time (longer or shorter).
Please use any theories, and show a role of unemployment in your model.
Please explain unemployment in a context of suicide.
Yes, You are right writting in disscusion : „Unemployment can lead to mental fatigue, a depressive state, or psychiatric disorders, all of which can lead to suicide” Yes it is a proces, and article should try to expalin data, not only compares in figures 1, 2.
Question 3. Realy? Did You find a sens of article, if You assume „ Our findings provide a starting point for discussions about economic-based risk factors for suicide based on statistical analyses using only numerical data without individual information. We did not examine the backgrounds of individual suicides.”
Ok, propobly You have not personal stories. But there are different analysis, publiactions, theories. Realy, are You able to cut off statistic and personal or sociala context?
Otherwise, should we take this triggers (as above) one by one and filter in countries’ statistics? But what for? Realy, our job will create new poins of view on suicide?
Question 4. Recomendation in conclusion – only skim the surface.
In my opinion it is effect sweeping research model and analysis.
On the other hand- without celar context of research, how to predict pragamtic conclusion? If we get a job for all, would suicidal problem among unemployed extincted?
Question 5. You predicted some differences between population (sex) but how to expalin it?
May it is only arificial effect.
In my country, in hard situation in 90., with structural economical problems, the rate of unemployed was enormous. Consequently, the stituation was dramatic among youth with no experience, and among women.
So, if I look for statistic (without context), the rate of suicide among unemployes persons, I can see negative tendecy. But sex is not a cause. Generally speaking women were stronger, and flexible in accommodation. The sacle was different.
Author Response
Reviewer 3:
This is an interesting study, shows social problems. But I don’t understand the main goal of study.
You decided to present correlational research. In your opinion joblessness is an interesting spectrum to observe fluctuation of suicidal rate.
Ok, but why unemployment? It is the same like different ones; heartbreak, impoverished relations, to be a victim of crime, homelessness, divorce, become widowed, debts and so on.
Following Your text we are witness of „figures playing”, unemployment – suicide - a few countries. The rates change in period of time, that’s all. It is not a new point of view.
That concept is quite obvious. In all countries, during structural, social and political changes, we are able to observe fluctuation of suicide.
Response: Thank you very much for your comments, which have helped us improve our manuscript. The revised text is presented in red (Reviewer 1), purple (Reviewer 2), and green (Reviewer 3) font in yellow highlighting.
We have addressed your comments below.
Question 1
Have you got any theoretical model of research? Accordance to Durkheim- there are facts (statistics), so what about argue.
Do you have any theoretical reason to take off „unemployment” as a specific (remarkable) factor for suicide? Generally speaking- are You able to explain your model of research in theories of suicide?
Response: We added text based on your suggestion in the 9th paragraph of the Discussion as follows.
“For example, Émile Durkheim’s theory of suicide focuses on a group-minded collective consciousness, and it examines four types of suicide (egoistic suicide, altruistic suicide, anomic suicide, and fatalistic suicide.) [48]. We speculate that our present findings may be linked to egoistic suicide in particular. It is also necessary to discuss new viewpoints based on the Durkheim theory in more precise studies of suicide in individual countries.”
Question 2.
Are You able to explain your point of view? How do You consider suicide as personal and social problem? Much as I admire your invention. But as far me that picture needs more knowledge about process of suicide (not as a suicide rate).
What is the first:
Do You think unemployment there is motive of suicide, or only a trigger of it? What is real cause?
A lot of theories have pointed out unemployment only as a trigger. It corresponds with personal features, patrimonial tendency.
Accordance my research (goodbye letters and personal story) there is a tendency:
Children- school and relations’ problems there are triggers of self-destructive behavior
Youth- love and heartbreak
Adults- joblessness, debts, divorce, bankruptcy
Old people- become widowed, loneliness feeling, physical suffering
Look, it changes in part of life, but the cause it means twine (web) of problems, and decision evolve in time (longer or shorter).
Please use any theories, and show a role of unemployment in your model.
Please explain unemployment in a context of suicide.
Yes, You are right writing in discussion : „Unemployment can lead to mental fatigue, a depressive state, or psychiatric disorders, all of which can lead to suicide” Yes it is a process, and article should try to explain data, not only compares in figures 1, 2.
Response: We added the following text based on your suggestion.
・In 1st paragraph of Introduction: “The main reasons that individuals attempt or commit suicide are personal problems, social problems, or combinations of both.”
・In last paragraph of Introduction: “Unemployment can lead to economic and life problems that contribute to motives for suicide, and unemployment has been associated with suicide in both Japan and South Korea [34,35].”
・In last paragraph of Introduction: “There are many triggers for suicide, and unemployment is a potential trigger that merits greater attention [36,37].”
・In 7th paragraph of Discussion: “The definition of ‘unemployed’ according to the ILO is ‘those who are older than a certain age, have no work in a specific period, can work during that period, and have been looking for a job in the most recent period’ [47]. Individual countries have defined ‘unemployed’ in accord with the ILO standard, but the content of the definitions differ slightly.”
Question 3.
Really? Did You find a sense of article, if You assume „ Our findings provide a starting point for discussions about economic-based risk factors for suicide based on statistical analyses using only numerical data without individual information. We did not examine the backgrounds of individual suicides.”
Ok, probably You have not personal stories. But there are different analysis, publications, theories. Really, are You able to cut off statistic and personal or social context?
Otherwise, should we take this triggers (as above) one by one and filter in countries’ statistics? But what for? Really, our job will create new points of view on suicide?
Response: In this study, we obtained only numerical data without individual information, and we contributed this report based on the statistical analyses of that data. We added the following text based on your comments.
In last sentence of the Conclusions section: “Suicide is a complex issue, and we believe that maintaining a wide perspective (including the content of this report) may be help identify new and effective suicide prevention measures.”
Question 4
Recommendation in conclusion – only skim the surface.
In my opinion it is effect sweeping research model and analysis.
On the other hand- without clear context of research, how to predict pragmatic conclusion? If we get a job for all, would suicidal problem among unemployed extinct?
Response: We added the following to the Conclusions section.
“If these entities and individuals become familiar with the present situation regarding the widespread increase in unemployment, they will be aware that (i) the rate of suicide may be increasing in their country, and (ii) many individuals in their country may be suffering from increased fatigue and poor physical and mental health. Such realizations could contribute to more effective suicide prevention measures. Suicide is a complex issue, and we believe that maintaining a wide perspective (including the content of this report) may be help identify new and effective suicide prevention measures.”
Question 5
You predicted some differences between population (sex) but how to explain it?
May it is only artificial effect.
In my country, in hard situation in 90., with structural economic problems, the rate of unemployed was enormous. Consequently, the situation was dramatic among youth with no experience, and among women.
So, if I look for statistic (without context), the rate of suicide among unemployed persons, I can see negative tendency. But sex is not a cause. Generally speaking women were stronger, and flexible in accommodation. The scale was different.
Response: There are reports with various viewpoints on this issue. The USSR collapsed in 1991 and separated into the 15 post-Soviet countries, and the present report was a pilot study of this issue in these countries. The 9th paragraph of the Discussion was revised as follows, and we plan to examined this in detail in the future based on Reviewer 1’s suggestion.
“Future investigations should apply additional statistical approaches including ecological analyses, and it is important to compare the existing findings with their results.”
Other
Response: The References list has been updated with new references and with ‘doi’ information.

Round 2
Reviewer 1 Report
I appreciate the thoroughness of the authors' responses to the reviewers' comments. I feel that the paper now better explains the findings (and their caveats and areas for future research).
Reviewer 3 Report
no comments in second round